# Critical Role of Caveolin-1 Loss/Dysfunction in Pulmonary Hypertension

**DOI:** 10.3390/medsci9040058

**Published:** 2021-09-22

**Authors:** Rajamma Mathew

**Affiliations:** Section of Pediatric Cardiology, Departments of Pediatrics and Physiology, New York Medical College, Valhalla, NY 10595, USA; rmathew@nymc.edu; Tel.: +1-914-594-4750

**Keywords:** caveolae, caveolin-1, endothelial cells, pulmonary hypertension

## Abstract

Pulmonary hypertension (PH) is a rare disease with a high morbidity and mortality rate. A number of systemic diseases and genetic mutations are known to lead to PH. The main features of PH are altered vascular relaxation responses and the activation of proliferative and anti-apoptotic pathways, resulting in pulmonary vascular remodeling, elevated pulmonary artery pressure, and right ventricular hypertrophy, ultimately leading to right heart failure and premature death. Important advances have been made in the field of pulmonary pathobiology, and several deregulated signaling pathways have been shown to be associated with PH. Clinical and experimental studies suggest that, irrespective of the underlying disease, endothelial cell disruption and/or dysfunction play a key role in the pathogenesis of PH. Endothelial caveolin-1, a cell membrane protein, interacts with and regulates several transcription factors and maintains homeostasis. Disruption of endothelial cells leads to the loss or dysfunction of endothelial caveolin-1, resulting in reciprocal activation of proliferative and inflammatory pathways, leading to cell proliferation, medial hypertrophy, and PH, which initiates PH and facilitates its progression. The disruption of endothelial cells, accompanied by the loss of endothelial caveolin-1, is accompanied by enhanced expression of caveolin-1 in smooth muscle cells (SMCs) that leads to pro-proliferative and pro-migratory responses, subsequently leading to neointima formation. The neointimal cells have low caveolin-1 and normal eNOS expression that may be responsible for promoting nitrosative and oxidative stress, furthering cell proliferation and metabolic alterations. These changes have been observed in human PH lungs and in experimental models of PH. In hypoxia-induced PH, there is no endothelial disruption, loss of endothelial caveolin-1, or enhanced expression of caveolin-1 in SMCs. Hypoxia induces alterations in membrane composition without caveolin-1 or any other membrane protein loss. However, caveolin-1 is dysfunctional, resulting in cell proliferation, medial hypertrophy, and PH. These alterations are reversible upon removal of hypoxia, provided there is no associated EC disruption. This review examined the role of caveolin-1 disruption and dysfunction in PH.

## 1. Introduction

Pulmonary hypertension (PH) is a serious disease with a high morbidity and mortality rate. A variety of systemic diseases and genetic mutations are known to be associated with PH. PH develops because of a deregulated response of pulmonary vasculature to injury induced by physical, chemical, inflammatory, or oxidative stress. It is a known sequel of a number of cardiopulmonary diseases including congenital heart defect, lung developmental disorders, chronic obstructive lung disease, other lung parenchymal diseases, collagen and autoimmune disorders, inflammatory diseases, and drug toxicity. In addition, some forms of genetic mutations render individuals prone to develop PH. Based on the underlying disease, PH is classified into five groups. Gr.1, designated as pulmonary arterial hypertension (PAH), includes idiopathic and heritable PAH and PAH associated with drug toxicity, inflammatory and autoimmune diseases, and congenital heart defects. Pulmonary capillary hemangiomatosis, pulmonary veno-occlusive disease, and persistent PH of the newborn are designated as Gr. 1′ and Gr. 1′′, respectively. The rest of the groups are labeled as PH. Included in the Gr. 2 is PH associated with left ventricular diseases, such as congenital or acquired left-heart inflow/outflow obstructive lesions and cardiomyopathies. Gr. 3 is comprised of alveolar hypoventilation disorders, chronic lung diseases, and lung developmental anomalies leading to PH. Included in Gr. 4 is chronic thromboembolic PH. PH associated with miscellaneous disorders such as myeloproliferative, hematological, thyroid, and renal diseases are incorporated in Gr. 5 [1]. Currently, mean pulmonary artery pressure of >20 mmHg and pulmonary vascular resistance of ≥3 Wood units are considered indicative of PAH [2]. PH is associated with altered vascular relaxation responses, pulmonary vascular remodeling, and obstruction, leading to increased pulmonary artery pressure, right ventricular hypertrophy, and subsequent right heart failure and premature death. Despite significant progress over the past 100+ years, the cure is still not in sight. Part of the problem is that the patients are diagnosed relatively late, after they have already developed significant histopathological changes in the pulmonary vasculature [3]. This is not surprising, because in animal experimental studies, the pulmonary arterial endothelial damage including the loss of membrane proteins and the activation of proliferative pathways has been shown to occur before the onset of PH [4]. The survival time in patients with PAH without treatment is reported to be 2.8 years [5]. Modern therapy has improved the quality of life and the survival rate; the 3-year survival rate in PAH patients is about 58–67% [6,7]; however, the vascular pathology progresses unabated [8]. Regardless of the underlying disease, endothelial cell (EC) disruption/dysfunction plays a major role in PH, leading to impaired availability of bioactive nitric oxide (NO), medial hypertrophy, and widespread proliferative changes in the pulmonary arteries [9]. ECs form a non-thrombogenic monolayer in contact with the underlying smooth muscle cells (SMCs) and with blood flow and mechanical forces on the other side. Juxtaposition of ECs and SMCs facilitates crosstalk; and ECs play a pivotal role in regulating vascular tone, permeability, inflammation, and coagulation, thus maintaining homeostasis [10]. The apoptosis rate in ECs is low under normal conditions. The luminal surface of ECs is coated with glycocalyx that forms an important barrier; it prevents platelet and leukocyte cells’ adhesion and modulates permeability. In addition, it mediates the shear-induced release of nitric oxide (NO) from ECs. In response to stress, endothelial glycocalyx is shed [11]. Heparan sulfate, a component of glycocalyx, mediates caveolar mechano-sensors. Interestingly, the expression of caveolin-1, a membrane protein is dependent on the heparan sulfate. Furthermore, disturbed flow has been shown to inhibit the expression of glycocalyx and caveolin-1, leading to the downregulation of mechano-signaling and reduced colocalization with serine 1177 phosphorylated endothelial-nitric oxide synthase (eNOS-pS1177) in caveolae [12]. Recent studies have shown that the loss or dysfunction of caveolin-1 plays a significant role in the initiation and the progression of the pathogenesis of PH.

## 2. Caveolae and Caveolin-1

Caveolae, flask-shaped, specialized lipid rafts (50–100 nm) found on the plasma membrane, were first described in the 1950s by Palade and Yamada [13,14]. Caveolae are present in a variety of cells including ECs, SMCs, epithelial cells, fibroblasts, and adipocytes. They form an important signaling platform that compartmentalizes and integrates a number of signaling molecules and allows cross talk between different signaling pathways [15]. Caveolae are thought to bud from the plasma membrane, a process regulated by the ATPase Eps-15 homology domain 2 (EHD2) to fuse with early endosomes. Caveolae play an important role in mechano-protection, signaling, and lipid regulation [16]. Several non-essential proteins are also involved in the biogenesis of caveolae biogenesis. PACSIN-2 is the only F-BAR protein representative of proteins regulating membrane curvature that has been involved in caveolae morphology. EHD2 localized at the neck of caveolae is thought to stabilize caveolae by controlling their dynamics in association with the actin cytoskeleton [17]. In addition, caveolae regulate Ca^2+^ signaling pathways present within caveolae, and the interaction of the store-operated Ca^2+^ entry channel with its modulators facilitates Ca^2+^ entry and modulates cellular functions [18].

Caveolin-1, a major protein (~22 kDa) constituent of caveolae that maintains the shape of caveolae, was identified in 1992 [19,20]. Caveolin-1 functions through protein–protein interaction, and it regulates and stabilizes several proteins including the Src family of kinases, G proteins (α-subunits), G protein-coupled receptors, H-Ras, protein kinase C (PKC), endothelial NO synthase (eNOS), integrins, and growth factor receptors such as vascular endothelial growth factor receptor (VEGF-R) 2 and epidermal growth factor receptor, and maintains them in an inhibitory conformation [21,22,23]. Caveolin scaffolding domain (CSD, amino acid 82–101) is an important domain for caveolin oligomerization and caveolae formation and is thought to be involved in the regulation of cellular signaling events [24].

Caveolin-1 has been shown to be a negative regulator of inter-endothelial junction permeability in vivo. Caveolin-1 loss increases constitutive endothelial permeability and reduces the levels of vascular endothelial cadherin (VE-Cadherin) and β-catenin levels. Furthermore, the loss of the endothelial barrier function is a significant phenomenon of inflammation [25,26]. Caveolin-1 plays a critical role in inflammation. Caveolin-1 knockout mice have been shown to exhibit increased IL-6, TNF-α, and IL-12p70 levels. Nonspecific stimulation increases cytokine production in circulating *CAV1*^−/−^ lymphocytes compared with the wild-type lymphocytes [27]. In addition, caveolin-1 is essential for the function of tissue repair processes, including adhesion, migration, and mechanical tension buffering; it inhibits major profibrotic signaling pathways and antagonizes profibrotic physiological events. Furthermore, low caveolin-1 expression accompanied by increased IL-1β and caspase-1 has been reported in alveolar epithelial type I cells from bleomycin-injured mouse lungs and in lung sections from patients with idiopathic pulmonary fibrosis. Overexpression of caveolin-1 is reported to suppress bleomycin-induced activation of caspase-1 and maturation of pro-IL-1β, both in mouse lungs and in primary type I epithelial cells [28]. 

Caveolin-1 regulates vascular tone. For optimal activation, eNOS is targeted to caveolae. Through its interaction, caveolin-1 inhibits eNOS; in response to various stimuli, such as shear stress and VEGF, eNOS binds to HSP90 in a Ca^2+^-calmodulin-dependent manner, thereby facilitating displacement of caveolin-1 from eNOS and increasing eNOS activity. However, caveolin-1 is crucial for agonist-induced eNOS activation. Caveolin-1 is thought to serve as an eNOS chaperon regulating NO production, independent of its localization within caveolae or its acylation state [29,30,31]. In addition to caveolae, caveolin-1 is also localized in mitochondria. The transfer of caveolin-1 between mitochondria and caveolae plays a significant role in the adaptation to cellular stress and injury. In response to shear stress, cells increase ATP production, leading to the Ca^2+^ release from intracellular Ca^2+^ stores. The resulting increase in the cytoplasmic Ca^2+^concentration leads to the increased production of NO, which plays a pivotal role in the regulation of blood flow-dependent vasodilation, blood pressure, and vascular remodeling. Importantly, caveolin-1 knockdown abrogates the shear stress-induced mitochondrial ATP generation, resulting in the loss of ATP release and influx of Ca^2+^ into the cells. Caveolin-1-mediated mechanisms appear to be essential for shear stress-induced activation of mitochondrial ATP in ECs [32]. Furthermore, caveolin-1 in SMCs regulates Ca^2+^ entry and enables vasoconstriction [33]. Caveolin-1 deficient mice exhibit impaired mitochondrial function. Caveolin-1 loss leads to free cholesterol accumulation in mitochondrial membranes, increased membrane condensation, and reduced efficiency of the respiratory chain and intrinsic antioxidant defense, resulting in increased susceptibility to apoptosis [34]. In addition, the loss of caveolin-1 has been shown to induce increased mitochondrial reactive oxygen species (ROS) and intracellular H_2_O_2_ production in ECs [35]; caveolin-1 reduces the production of ROS through inhibiting the NF-kB pathway [36]. 

Transforming growth factor (TGF) βs are multifunctional growth factors that regulate a number of physiological processes such as embryonic development, stem cells, and tissue homeostasis and repair. Deregulation of TGF-βs leads to pathological processes such as cancer and cardiovascular diseases [37]. TGF-β components are associated with caveolin-1, and caveolin-1 suppresses TGF-β-mediated phosphorylation of Smad2 and the downstream events [38]. Furthermore, perturbation of caveolin-1 function results in deregulation of TGF-β signaling pathways, leading to increased collagen deposition and alterations in the composition of extracellular matrix, contributing to increased resistance [39]. These results underscore the significance of caveolin-1’s role in maintaining vascular health. 

Caveolin-2 is another protein expressed in several cell types including ECs and SMCs, and it co-localizes with caveolin-1; it requires caveolin-1 for its transport from the Golgi body to the plasma membrane. The absence of caveolin-1 results in the degradation of caveolin-2, and its loss facilitates cell proliferation [40,41]. However, caveolin-2 is not necessary for caveolar localization of caveolin-1; but the co-expression of caveolin-1 and -2 results in a more efficient formation of caveolae [42]. Polymerase 1 and transcript release factor (PTRF), also known as cavin-1, is an essential component of caveolae; it regulates membrane curvature by stabilizing caveolin-1 in caveolae. The loss of cavin-1 results in the loss of caveolae and the release of caveolin-1 into the plasma membrane. Furthermore, caveolin-1 is required for cavin-1 recruitment to plasma membrane, and cavin-1 is essential for caveolae formation and caveolin-1 stabilization [43,44]. Importantly, caveolin-1 loss is accompanied by a marked loss of caveolin-2 and partial reduction in cavin-1 expression in the lungs. The re-expression of caveolin-1 rescues and stabilizes caveolin-2 and cavin-1 [45]. Cavin-1^−/−^ mice have been reported to display lung pathological changes such as remodeled pulmonary vessels, PH, and RVH [46]. In addition, cavin-1 loss has been shown to promote neointima formation; the overexpression of cavin-1 suppresses vascular SMC proliferation and migration, whereas its inhibition promotes cell proliferation and migration [47]. These results highlight the interrelationship between caveolin-1 and cavin-1 in maintaining homeostasis. 

## 3. Caveolin-1 Expression and Function in Pulmonary Hypertension and Caveolin-1

Austin et al. [48] reported a frameshift mutation in *CAV1*, P158PfsX22, associated with reduced expression of caveolin-1 protein in a patient with idiopathic PAH. Immunostaining of the lung tissue revealed significant reduction in caveolin-1 expression in the pulmonary arterial ECs. Furthermore, P158PfsX22 frameshift was thought to introduce a gain of function, giving rise to a dominant negative form of *CAV1* [49]. *CAV1*^−/−^ mice are viable, but they develop dilated cardiomyopathy and PH, which are attenuated by caveolin-1 re-expression [50,51]. It was further shown that persistent eNOS activation in caveolin-1^−/−^ lungs results in tyrosine nitration of protein kinase G (PKG) and impairment of its activity, leading to PH. Furthermore, eNOS activation and PKG nitration concomitant with caveolin-1 deficiency was observed in lungs from patients with idiopathic PAH [52]. Interestingly, a membrane-permeable chimeric peptide containing a cellular internalization sequence fused to the caveolin-1 scaffolding domain (residues 82–101) has been demonstrated as a potent inhibitor of eNOS in mice and to reduce inflammation [53]. Furthermore, Loss and dysfunction of endothelial caveolin-1 without associated genetic mutation have been observed in human and experimental PH. The loss of endothelial caveolin-1 in PH is progressive and is accompanied by increased inflammatory and proliferative pathways. The progressive disruption of EC and endothelial caveolin-1 loss are accompanied by an increase in caveolin-1 expression in SMC and subsequent neointima formation. 

## 4. Endothelial Cell Disruption and Caveolin-1 Loss

Conditions such as inflammation, shear stress, increased pulmonary blood flow and pressure, and drug toxicity are known to induce disruption of ECs, resulting in the loss of endothelial membrane proteins including caveolin-1. The loss of endothelial caveolin-1 is accompanied by increased expression of cytokines and pro-proliferative and anti-apoptotic pathways, leading to PH [11,54,55]. In addition, increased serum levels of IL-1 β and IL-6 were observed in severe PAH, indicating a role for proinflammatory cytokines in PAH. Furthermore, high levels of cytokines, chemokines, and inflammatory mediators detected in PAH patients correlate with worse clinical outcome [56,57]. Monocrotaline (MCT)-induced PH in rats has been reported to be associated with an inflammatory response, as shown by an early and progressive increase in IL-6 mRNA expression and IL-6 bioactivity in the lungs [58]. Furthermore, ectopic upregulation of membrane-bound IL-6 receptor (IL6R) on pulmonary arterial SMCs in PAH patients and in rodent models of PH has recently been reported. In an animal model, deletion of IL6R in SMC was shown to prevent the development of PH [59]. IL-6 is known to activate signal transducers and activators of transcription 3 (STAT3), a transcription factor that positively regulates cell growth and proliferation. Furthermore, persistent activation of PY-STAT3 has been reported in the MCT model of PH [60] and in pulmonary arterial ECs from patients with idiopathic PAH [61]. Importantly, caveolin-1 has been shown to act as a suppressor of cytokine signaling and to inhibit PY-STAT3 activation [62]. Interferon (IFN) α and β are currently in use for various hematological disorders, cancer, and infectious diseases. IFN therapy, however, has been shown to be associated with vasculopathy, and it is thought that IFN pathway may have a role in the pathobiology of PH [63]. A recent study has shown that the *Cav1*^−/−^ mice exhibit activation of STAT1 and AKT in lungs accompanied by increased circulating levels of CXCL10, indicative of IFN-mediated inflammation. Exogenous IFN was found to reduce caveolin-1 expression and activate STAT1 and AKT. In addition, it altered the cytoskeleton of PAECs. Furthermore, PAH patients with *CAV1* mutations exhibit elevated serum CXCL10 levels [64]. These results indicate that inflammation plays a significant role in PH, and, importantly, caveolin-1 plays a critical role in modulating inflammatory responses. 

The loss of endothelial caveolin-1 was first observed in MCT-induced PH in rats [60] and myocardial infarction-associated PH in rats [65]. MCT is a well-established model of PH. A single sc injection leads to the disruption of EC membrane associated with a progressive loss of caveolin-1 and other membrane proteins such as PECAM-1, Tie2, soluble guanylate cyclase, and VE-Cadherin. In addition, progressive activation of proliferative pathways such as PY-STAT3, Bcl-xl, and pERK, and increasing expression and activation of matrix metalloproteinase 2 (MMP2) has been reported. These alterations are observed before the development of PH. The loss of caveolin-1, PECAM-1, and VE-Cadherin is indicative of disruption of endothelial membrane integrity and barrier function. At 2 weeks post-MCT, extensive loss of caveolin-1 is accompanied by a loss of proteins such as HSP 90, Akt, and Iκ-β IκB-α, indicative of EC cytosolic damage and PH. However, at this stage, significant loss of endothelial caveolin-1 accompanied by relatively well-preserved eNOS leads to eNOS uncoupling and the generation of reactive oxygen species (ROS). By 4 weeks post-MCT, a significant loss of eNOS results in normalization of ROS production. At this stage, there is a significant loss of vWF, indicative of extensive EC disruption that is accompanied by an enhanced expression of caveolin-1 in SMCs. Importantly, the loss of endothelial caveolin-1 accompanied by enhanced expression of caveolin-1 in SMCs has been observed in the lungs of PAH patients [66,67,68].

Oliveira et al. [69] recently showed the loss of endothelial caveolin-1 accompanied by increased plasma concentration of caveolin-1, extracellular vesicles, and increased expression of TGF-β in Sugen + hypoxia model of PH. Caveolin-1 loss appears to be a key factor in TGF-β-induced pulmonary vascular remodeling. In addition, lungs from patients with acute respiratory distress syndrome display reduced endothelial caveolin-1, increased TGF-β levels, reduction in bone morphogenic protein receptor type 2 (BMPR2) levels, and PH. Endothelial caveolin-1 depletion and oxidative stress result in a reduction in BMPR2 expression and enhanced TGF-β-driven SMAD-2/3 signaling, thus, promoting pulmonary vascular remodeling in inflamed lungs [70]. The BMPR2 gene belongs to the TGF-β family. Heterozygous germline mutations in the BMPR2 gene have been identified as critical in the development of PAH [71]. In addition, BMPR2 loss occurs in PAH patients without any evidence of BMPR2 mutation [72]. BMPR2 loss has also been observed in the MCT and hypoxia models of PH [73]. Importantly, the reduced BMPR2 expression induces endothelial mitochondrial dysfunction and promotes pro-inflammatory and pro-apoptotic states. Furthermore, BMPR2-deficient mice develop hypoxia-induced PH that is not reversed upon reoxygenation, unlike the wild-type mice [74]. In addition, pulmonary artery SMCs from BMPR2 (+/−) mice and from patients with BMPR2 mutations have been shown to produce higher levels of IL-6 and KC/IL-8 when exposed to lipopolysaccharide stimulation compared with the controls. These cells exhibit increased phospho-STAT3 and a loss of extracellular superoxide dismutase [75]. Increased expression of several cytokines has been observed in patients with PAH. Increased expression of CCL5, also known as RANTES, was reported in lung samples from patients with severe PAH. Importantly, ECs are the major source of CCL5 [76]. Interestingly, in Sugen + hypoxia model of PH in mice, deletion of CCL5 significantly attenuated PH by restoring caveolin-1-dependent amplification of BMPR2 signaling. In the cell culture studies, CCL5 deficiency increased apoptosis and tube formation of pulmonary arterial ECs and suppressed proliferation and migration of pulmonary arterial SMCs [77]. In a different study, elafin, an elastase inhibitor was shown to reverse obliterative changes in pulmonary arteries via caveolin-1-dependent amplification of BMPR2 signaling in Sugen + hypoxia-induced PH in rats. In addition, in cell culture studies, elafin promoted angiogenesis by increasing pSMAD-dependent and -independent BMPR2 signaling in pulmonary arterial ECs from normal and PH patients [78]. In normal lungs, BMPR2 is located within lipid-dense fractions of the pulmonary EC membrane with a portion present in caveolae, suggesting a potential dynamic regulatory structural relationship with caveolin-1 [79] These results underscore the importance of caveolin-1 and BMPR2 interaction in maintaining homeostasis.

In addition, caveolin-1 loss in the MCT model is accompanied by a loss of caveolin-2 and cavin-1 [67,80]. In all cell types, caveolin-2 is present in association with caveolin-1; it requires caveolin-1 for its transport from Golgi body to the plasma membrane. The absence of caveolin-1 results in the degradation of caveolin-2, and its loss facilitates cell proliferation [40,41].

### 4.1. Enhanced Expression of Caveolin-1 in SMC

Endothelial apoptosis has been shown to play a key role in stimulating SMC growth [81], further supporting the significant role of ECs in maintaining homeostasis. At 4 weeks post-MCT, about 29% of the arteries show a loss of vWF, accompanying endothelial caveolin-1 loss. Importantly, 70% of these arteries with vWF loss exhibit enhanced expression of caveolin-1 in SMCs [66]. The loss of vWF accompanying endothelial caveolin-1 loss is indicative of extensive EC damage and/or loss, which would expose SMC to increased shear stress and strain because of the elevated pulmonary artery pressure that may facilitate increased expression of caveolin-1 in SMCs. In cell culture studies, pulmonary artery SMCs from patients with idiopathic PAH have been shown to express more caveolin-1 that contributes to increased capacitive Ca^2+^ entry and DNA synthesis, which could be reversed by silencing caveolin-1 [68]. The loss of ECs exposes underlying SMCs to direct pressure and shear stress. Caveolae, the plasma membrane sensors, flatten in response to membrane stretch. The flattening is a protective mechanism that buffers the membrane and prevents its rupture. However, the flattening of the membrane leads to caveolin-1 and cavin-1 dissociation [82]. Furthermore, in a balloon-injury model of neointima formation, cavin-1 loss was shown to promote neointima formation; the overexpression of cavin-1 suppressed vascular SMC proliferation and migration [47].

Vascular SMCs are unresponsive to mitogens under normal tensile stress. During altered mechanical stress, in response to growth factors, protein synthesis in these cells is upregulated, resulting in cell proliferation and neointima formation. Furthermore, in cell cyclic strain, caveolin-1 is thought to be critically involved in proliferation signaling, stretch-induced activation, and cell cycle entry [83]. In addition to the enhanced expression, caveolin-1 in SMCs is tyrosine phosphorylated [80]. Importantly, phosphorylated caveolin-1 in cancer cells has been shown to facilitate cell migration [84]. These studies indicate that this enhanced expression of caveolin-1 in SMCs is a serious sequela of EC loss and may be an important alteration that facilitates neointima formation, leading to the irreversibility of PH. The role of caveolin-1 appears to be dependent on its location (caveolar or non-caveolar site), the stimulus the cells are exposed to, and its location in caveolar or non-caveolar site.

### 4.2. Caveolin-1 Loss in Neointimal Cells and Plexiform Lesions

Endothelial apoptosis results in apoptosis-resistant, phenotypically altered cell proliferation, leading to severe PH. Furthermore, it has been shown that apoptotic cells are not present in plexiform lesions in the lungs of PAH patients [85]. Importantly, the overexpression of EC growth factors such as vascular endothelial growth factor VEGF and angiopoietin-1 have been shown to prevent the development of MCT-induced PH [86,87]. In addition, inhibition of VEGF receptor 2 accompanied by hypoxia has been shown to lead to EC proliferation and severe PH. It is thought that the increased apoptosis of ECs in response to the loss of survival signaling creates conditions that favor the emergence of apoptosis-resistant cells with increased growth potential [88]. Furthermore, neointimal cells in lungs from patients with idiopathic PAH and in MCT + hypoxia model of PH in rats show positive staining for SMC α-actin and scant caveolin-1 expression but normal eNOS expression [67], which is likely to result in eNOS uncoupling, leading to oxidative and nitrosative stress. The plexiform lesions in primary pulmonary hypertension are considered to be the result of an abnormal growth of modified smooth muscle cells (SMCs), perivascular inflammatory cells, and deregulated growth of endothelial cells [89]. Importantly, strong eNOS and low cav-1 expression have been reported in the neointimal and plexiform lesions [90,91]. In addition, ECs from patients with idiopathic PAH do show caveolin-1 degradation induced by sustained nitric oxide production. An increase in caveolin-1 mRNA expression indicates that the decrease in caveolin-1 expression occurs at the protein level [92]. The major cause of PH in caveolin-1 knockout mice is the uncoupling of eNOS and resulting oxidative stress and PH that is reversed by restituting caveolin-1 or by blocking eNOS [51,93].

In addition, the loss of caveolin-1 has been shown to alter mitochondrial function. Loss of caveolin-1 results in the accumulation of cholesterol in mitochondrial membrane, leading to mitochondrial dysfunction and alterations in metabolism [94]. In vitro studies with breast cancer cells have revealed that the loss of caveolin-1 results in the activation of NF-E2-related factor 2 (Nrf2), a transcription factor upstream of MnSOD, leading to an increase in the glycolytic rate, dependent on mitochondrial H_2_O_2_ production and AMPK activation. Rescue of caveolin-1 expression suppressed Nrf2 and reduced MnSOD expression. Furthermore, low caveolin-1 and high MnSOD levels are associated with a poor prognosis [95]. Endothelial mesenchymal transition (EndMT) is a process by which ECs exhibit phenotypic alteration and acquire the properties of myofibroblasts or mesenchymal cells. These cells exhibit loss of PECAM-1 and VE-Cadherin in addition to caveolin-1 and express SM α-actin. Barrier function and tight junction stability are lost. These transformed ECs acquire a pro-inflammatory phenotype and are primed for proliferation, migration, and tissue generation [96]. Importantly, caveolin-1 deficiency has been shown to induce spontaneous EndMT in pulmonary EC in vitro [97]. These studies indicate that the loss of caveolin-1 in neointimal cells and plexiform lesions results in cell proliferation and metabolic and mitochondrial dysfunction, leading to the irreversibility of PH.

The result of EC injury and pathways leading to irreversible PH are depicted in Figure 1. 

## 5. Endothelial Caveolin-1 Dysfunction without Endothelial Disruption or Loss

Hypoxia plays an important role in the pathogenesis of PH. Exposure to acute hypoxia results in pulmonary arterial contraction and elevated pulmonary artery pressure, whereas sustained hypoxia leads to pulmonary vascular remodeling, medial thickening, and extension of SMCs into partially muscular arteries [98]. Hypoxia alters the physical state of EC, lipid composition, and plasma membrane function. These alterations are reversible on return to a normoxic state [99]. Hypoxia impairs endothelium-dependent relaxation; however, endothelium-denuded vessels respond normally to nitroprusside. This suggests that hypoxia causes endothelial dysfunction; however, the relaxation response of SMCs is not affected [100,101]. Murata et al. [102] have shown that in pulmonary arteries from rats exposed to hypoxia for 1 week, eNOS forms a tight complex with caveolin-1 and becomes dissociated from HSP90 and calmodulin, leading to eNOS dysfunction. Furthermore, bovine pulmonary arterial ECs exposed to 24 h of hypoxia exhibit a complex formation between eNOS and caveolin-1 complex formation and the activation of PY-STAT3 [54]. The eNOS and caveolin-1 complex formed during hypoxia renders both molecules dysfunctional. 

Recent studies have shown that rats and bovine calves, when exposed to hypoxia, develop PH and SMC hypertrophy. Unlike the MCT model, there was no evidence of EC disruption or loss of endothelial caveolin-1, vWF, eNOS, or HSP90. Since there was no EC disruption, it was not surprising that there was no enhanced expression of caveolin-1 in SMCs. However, despite the presence of endothelial caveolin-1, there was a significant loss of phosphatase and tensin homolog [PTEN] and an increased expression of PY-STAT3 and β-catenin, indicating dysfunctional caveolin-1 [103]. PTEN negatively regulates STAT3 activation and β-catenin localization [104,105]. Furthermore, PTEN contains a caveolin-1 binding motif and, in part, co-localizes with it in caveolae [106]. Caveolin-1 expression is necessary for the membrane localization of PTEN. Fibroblasts from idiopathic pulmonary fibrosis lungs exhibit low caveolin-1 levels accompanied by low membrane PTEN levels; the overexpression of caveolin-1 has been shown to restore PTEN expression [107]. In addition, chronic hypoxia reduces ATP-induced Ca^2+^ entry in pulmonary endothelial cells, which can be resolved by introducing caveolin-1 scaffolding domain [108]. These results suggest that normally functioning caveolin-1 is essential for maintaining homeostasis. 

Mild cases of chronic obstructive pulmonary disease (COPD) exhibit endothelial dysfunction, and the loss of endothelium-dependent vascular relaxation correlates with the severity of the disease [109]. Interestingly, loss of endothelial caveolin-1 accompanied by enhanced expression of caveolin-1 in SMCs has been reported in COPD associated with PH. In contrast COPD without PH exhibited preserved endothelial caveolin-1 [110]. In addition, plexiform and angiomatoid lesions, like what is observed in idiopathic PAH, have been documented in explanted lungs obtained after transplantation in COPD associated with severe PH [111]. In infants with respiratory distress syndrome, despite PH and significant medial thickening, the expression of endothelial caveolin-1, PECAM-1, and vWF in pulmonary arteries was reported as well preserved, without any evidence of EC disruption or enhanced expression of cav-1 in SMC. However, disruption of ECs and endothelial caveolin-1 loss accompanied by enhanced expression of caveolin-1 in SMCs was observed in infants with bronchopulmonary dysplasia, associated with inflammation and PH [112]. Endothelial caveolin-1 is well preserved during hypoxia-induced PH, indicating that the EC membrane integrity is not disrupted. Thus, the preserved EC integrity may explain the reversal of hypoxia-induced PH on upon return to normoxic conditions (Figure 2). However, if endothelial disruption occurs, it would lead to the loss of endothelial caveolin-1, enhanced expression of caveolin-1 in SMCs, and the disease progression toward the irreversible stage.

Therapeutic Considerations: Caveolin-1 functions as a gatekeeper. It interacts with a number of transcription factors and regulates cell proliferation, cell migration, oxidative stress, inflammation, metabolism, and mitochondrial function and maintains homeostasis. However, caveolin-1 needs to be in caveolae for its proper function. Cavin-1 colocalizes with caveolin-1, maintains caveolar structure, and stabilizes caveolin-1 in caveolae. When caveolae are flattened, as a result of shear stress, caveolin-1 appears on the plasma membrane and it loses its protective function and becomes pro-proliferative and pro-migratory. The dual role of caveolin-1 is an important point to be considered when designing therapeutic measures. In experimental models of PH, early treatment as a preventive measure with several different drugs has provided positive results. These drugs, when administered early, probably protect the cell membrane integrity and endothelial caveolin-1, thus attenuating the disease progression. However, once PH is established, the therapeutic measures have been unsuccessful in reversing or halting the disease progression. Based on the data available about the caveolin-1 alterations in PH, one could postulate a few possible therapeutic measures such as restitution of endothelial caveolin-1 and inhibition of SMC-specific caveolin-1 to counteract its pro-proliferative function.
Restitution of endothelial caveolin-1: In experimental models the use of cavtratin has been found to be effective during the early phase of PH. Our preliminary studies show that cavtratin may also attenuate the progression of PH. Male Sprague–Dawley rats (*n* = 3–4) were treated with MCT (40 mg/kg) and exposed to hypoxia for 3 weeks, and then started on cavtratin or control peptide (2.5 mg/kg) ip every day for the next 2 weeks, while still being maintained in hypoxia. At the end of 2 weeks of treatment with cavtratin, the rats were studied. The pulmonary artery pressure (mmHg, Controls, 20 ± 0, MCT + hypoxia 95 ± 5 *, MCT + hypoxia + Cavtratin 57 ± 1 *#) and RV/LV ratio (Controls, 0.24 ± 0.02, MCT + hypoxia, 0.72 ± 0.05 *, MCT + hypoxia + Cavtratin 0.54 ± 0.02 *#; * = *p* < 0.05 vs. control, # = *p* < 0.05 vs. MCT + hypoxia) were significantly reduced in the cavtratin-treated groups and pro-migratory function. As shown in Figure 3, cavtratin treatment resulted in slightly less thickening of the pulmonary arteries (A) and preserved endothelial caveolin-1 (B).

Restitution of endothelial caveolin-1 after the development of PH does appear to attenuate the progression of the disease. This caveolin-1 should be able to prevent eNOS uncoupling, oxidative stress, and the activation of pro-proliferative and pro-migratory pathways. However, the effects of long-term therapy with caveolin-1 restitution needs to be evaluated to see whether it could reverse the disease process. Furthermore, cavin-1 and caveolin-1 work in tandem, and cavin-1 is essential for caveolin-1 stabilization in caveolae. Therefore, in addition to the restitution of caveolin-1, introduction of cavin-1 might be beneficial.
2.Inhibition of SMC-specific caveolin-1: The enhanced expression of caveolin-1 in SMC has been reported in IPAH as well as in experimental models. This enhanced expression appears to occur following the extensive endothelial disruption and the loss of endothelial caveolin-1. Patel et al. [68] were the first ones to report the presence of enhanced expression of caveolin-1 in IPAH. They have further shown that the isolated SMCs from IPAH patients exhibit not only enhanced expression of caveolin-1 but also increased capacitive Ca^2+^ entry and DNA synthesis, which could be reversed by blocking caveolin-1. This is an important observation. Furthermore, this caveolin-1 appears to be phosphorylated [79], and p-caveolin-1 has been shown to be pro-migratory [87]. These results indicate that the inhibition of SMC caveolin-1 using SMC-specific caveolin-1 antibody might be warranted.

There is a theoretical possibility that, by restituting endothelial caveolin-1 and inhibiting enhanced expression of caveolin-1 in SMCs, the progression of PH may be halted or may even be reversed. Restitution of caveolin-1 in neointimal cells would be able to inhibit oxidative/nitrosative stress induced by the uncoupling of eNOS, improving metabolic deregulation and mitochondrial stress. The addition of cavin-1 in combination with endothelial caveolin-1 may have a better result. However, more work needs to be done to explore the ways to improve endothelial caveolin-1 expression and function, preservation of caveolae, and inhibition of enhanced expression of caveolin-1 in SMCs.

## Figures and Tables

**Figure 1 medsci-09-00058-f001:**
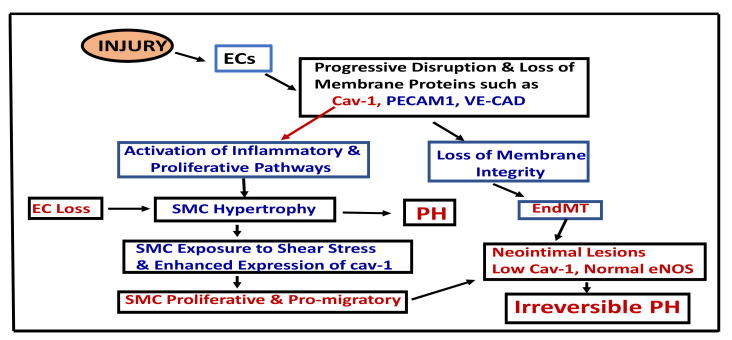
This figure depicts the result of EC damage, loss of endothelial membrane and pathways, leading to irreversible PH. Cav-1 (caveolin-1), ECs (endothelial cells), EndMT (endothelial mesenchymal transition), PECAM-1 (Platelets’ endothelial cells’ adhesion molecules), PH (pulmonary hypertension), SMC (smooth muscle cells), VE-CAD (vascular endothelial cadherin).

**Figure 2 medsci-09-00058-f002:**
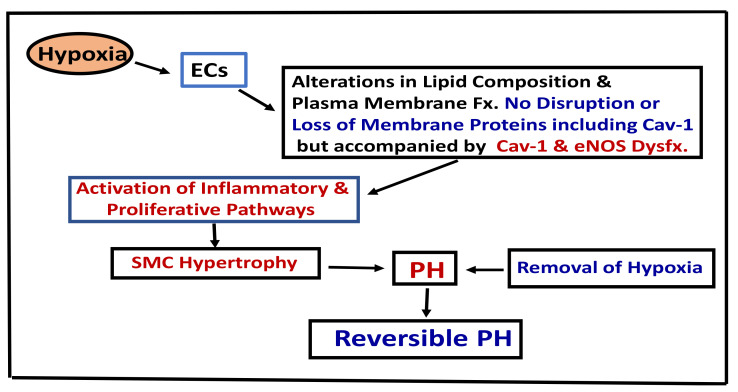
This figure depicts the pathways leading to hypoxia-induced PH. Cav-1 (Caveolin-1), Fx (function), Dysfx (dysfunction), ECs (endothelial cells), PH (pulmonary hypertension), SMC (smooth muscle cells).

**Figure 3 medsci-09-00058-f003:**
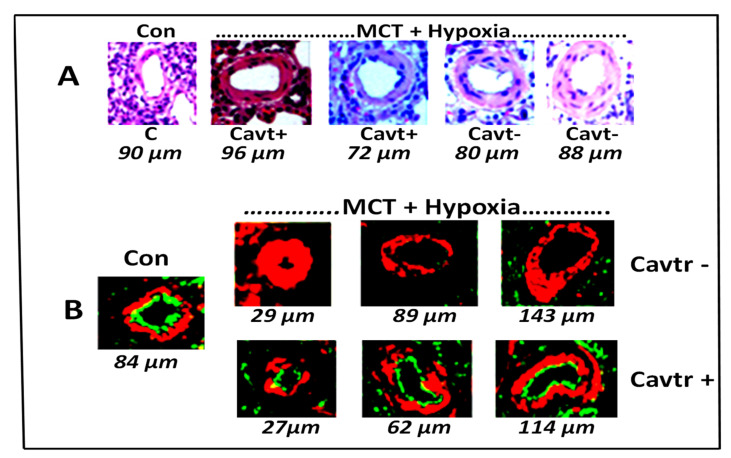
(**A**) Pulmonary artery (size 72–96 µm) histology in C (controls), Cavtr+ (M + H+ cavtratin) and Cavtr– (M + H). The artery in the control is thin walled. In the M + H group the arteries appear thickened. In the cavtratin-treated group, the arteries appear less thickened. (**B**) Immunofluorescence study: Caveolin-1 (green) and smooth muscle α-actin (red) in pulmonary arteries (size 27–143 µm). There is significant loss of endothelial caveolin-1 in MCT + hypoxia group, and cavtratin-treated rats show recovery of endothelial caveolin-1.

## Data Availability

Not applicable.

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
