# Peer review of "Critical Role of Caveolin-1 Loss/Dysfunction in Pulmonary Hypertension"

_medsci, 2021, doi:10.3390/medsci9040058_

Round 1

Reviewer 1 Report

Authors review the literature concerning caveolin-1 and pulmonary hypertension.

The organization of this manuscript is not clear. Are sections with the numbers under “Pulmonary Hypertension and Caveolin-1” paragraph? The role of this paragraph is not clear.

There should be a more distinct Summary section.

What data support the sentences such as “These alterations facilitate irreversibility of PH” and “Thus, caveolin-1 plays a very important role in the pathogenesis of PH.”?

Please provide some figures.

Please provide some insights to how understanding caveolin-1 in PH help develop therapeutic agents to prevent and/or treat PH.

Please provide future perspectives in this research field.

While the title of this review article is “Critical Role of Caveoline-1 in Pulmonary Hypertension”, much of the information provided are concerning changes in the caveolin-1 levels without providing the information on causal links.

Author Response

Authors review the literature concerning caveolin-1 and pulmonary hypertension. The organization of this manuscript is not clear. Are sections with the numbers under “Pulmonary Hypertension and Caveolin-1” paragraph? The role of this paragraph is not clear. I apologize for the confusion. The heading has been changed to “Caveolin-1 Expression and Function in Pulmonary Hypertension”. The following paragraphs deal with the disruption of endothelial caveolin-1 and subsequent alterations and endothelial caveolin-1 dysfunction without disruption or loss. I hope the present description is clearer.

 There should be a more distinct Summary section. The summary section has been changed.

 What data support the sentences such as “These alterations facilitate irreversibility of PH” and “Thus, caveolin-1 plays a very important role in the pathogenesis of PH.”? I have changed these sentences to avoid confusion.

Please provide some figures. I have added 3 figures.

 Please provide some insights to how understanding caveolin-1 in PH help develop therapeutic agents to prevent and/or treat PH. I have included a section on “Therapeutic Considerations” and our preliminary results obtained after 2 weeks cavtratin treatment started at 3 weeks post-MCT + hypoxia. 

 Please provide future perspectives in this research field. I have included a section at the end regarding the future perspectives in the PH field.

While the title of this review article is “Critical Role of Caveoline-1 in Pulmonary Hypertension”, much of the information provided are concerning changes in the caveolin-1 levels without providing the information on causal links. I have tried my best to show the importance of caveolin-1 role in PH.  

Reviewer 2 Report

Minor Revision is required:

The review article by Rajamma Mathew summarizes the role of caveolin-1 in pulmonary hypertension. This review is well written and provides a comprehensive overview of recent research papers on endothelial cell dysfunction, endothelial disruption related with caveolin-1 loss as well as decrease in caveolin-1 in plexiform lesions and neointimal lesions. The authors focus on the perturbation of different cellular signaling as a result of  caveolin-1 loss, which could be informative to develop drugs for clinical trials.

Major comments:
1. Some of the sections can be better structured rather than just listing papers regarding caveolin-1 and endothelial dysfunction. Possibly, some selected papers could be discussed in more detail, and unanswered questions should be highlighted to start to help the readers to understand where the field is currently standing and where it should go.

2.There are new evidence on constitutive activation of  interferon signaling with caveolin1 loss. Type I interferon activation and endothelial dysfunction in caveolin-1 insufficiency-associated pulmonary arterial hypertension (PMID: 33836561).This recent study supports the idea that caveolin-1 have fundamental role in regulating inflammatory response and should be included in Endothelial Cell Disruption & Caveolin-1 Loss section of the paper.

3.It would be helpful to include  information on the clinical trials that target Caveolin1 loss associated signaling. For example, possible therapeutic approaches could be discussed (e.g., blocking antibodies, blocking peptides, small molecules)

Author Response

Rev #2

Minor Revision is required:

The review article by Rajamma Mathew summarizes the role of caveolin-1 in pulmonary hypertension. This review is well written and provides a comprehensive overview of recent research papers on endothelial cell dysfunction, endothelial disruption related with caveolin-1 loss as well as decrease in caveolin-1 in plexiform lesions and neointimal lesions. The authors focus on the perturbation of different cellular signaling as a result of caveolin-1 loss, which could be informative to develop drugs for clinical trials. Thank you for your comments.

Major comments:
1. Some of the sections can be better structured rather than just listing papers regarding caveolin-1 and endothelial dysfunction. Possibly, some selected papers could be discussed in more detail, and unanswered questions should be highlighted to start to help the readers to understand where the field is currently standing and where it should go.   I have tried to be clearer about the role of caveolin-1 in PH and the need for studies dealing with the restitution of normally functioning endothelial caveolin-1 and inhibition of abnormal caveolin-1 expression in SMCs.

  1. There are new evidence on constitutive activation of interferon signaling with caveolin1 loss. Type I interferon activation and endothelial dysfunction in caveolin-1 insufficiency-associated pulmonary arterial hypertension (PMID: 33836561).This recent study supports the idea that caveolin-1 have fundamental role in regulating inflammatory response and should be included in Endothelial Cell Disruption & Caveolin-1 Loss section of the paper. I apologize for the oversight. I have included the paper you have mentioned and also the one that appeared in 2014 regarding interferon. Savale L et al.  Pulmonary arterial hypertension in patients treated with interferon. Eur Respir J. 2014; 44:1627-1634.  
  2. It would be helpful to include information on the clinical trials that target Caveolin1 loss associated signaling. For example, possible therapeutic approaches could be discussed (e.g., blocking antibodies, blocking peptides, small molecules). I have included the possibility of restitution of endothelial caveolin-1 and inhibition of enhanced caveolin-1 expression in SMCs as therapeutic measures.

Round 2

Reviewer 1 Report

This paper is not well organized.

The paper has subheadings:

Introduction

Caveolae and Caveolin-1

Caveolin-1 Expression and Function in Pulmonary Hypertension

1. Endothelial Cell Disruption & Caveolin-1 Loss

1a. Enhanced Expression of Caveolin-1 in SMC

1b. Caveolin-1 Loss in Neointimal Cells and Plexiform Lesions

2. Endothelial Caveolin-1 Dysfunction without Endothelial Disruption or Loss

1. Restitution of endothelial caveolin-1

2. Inhibition of SMC-spedific caveolin-1

Why is the numbering system like this?

It is unclear if the last paragraph is a part of the section “2. Inhibition of SMC-specific caveolin-1” or is a paragraph that serves as the Summary/Conclusions of the whole paper. Please have distinct “Conclusions” section as well as some statement on future perspective.

Figure 2: What does the author mean by “Loss of membrane proteins including Cav-1 but accompanied by Cav-1 dysfunction”? If the protein is lost, how can it be dysfunctioned?

Figure 3 is noted in the text.

While Figure 3 has panels A and B, the legend does not describe A and B.

Labels in Figure 3 are not legible.

Fig. 3 legend: What is “M + H”?

Fig. 3 legend: “In he covtratin treated group Table. (green) and smooth muscle alpha-actin (red).” is not a sentence.

Fig. 3: What are 29, 89, 143, 27, 62 and 114 μm?

Author Response

Response to the Reviewer:

The author needs to improve the resolution for "Figure 3. A: Pulmonary artery (size 72-96 μm) histology in C (controls), Cavtr+ (M + H+ cavtratin) and Cavtr – (M + H) to meet the production standard (1200 dpi). Scale bars should be provided for all the panels.

Thank you for your comments. The figure 3 has been improved. The pictures were taken at 20X or 40x magnification depending on the size of the artery to obtain clear depiction. Adding scale bars would make the slide too crowded. I have included the sizes of different arteries. I hope it is acceptable now.

I have made following changes. The added words appear green in the text.

  1. 1. Page 5, under the heading “ Endothelial Cell Disruption & Caveolin-1 Loss:” line 9: the words “in rats” has been added for clarity, as shown in the following statement. .

          “Monocrotaline (MCT)-induced PH in rats has been reported to be associated with an inflammatory response as shown by an early and progressive increase in IL-6 mRNA expression and IL-6 bioactivity in the lungs [58].”  

  1. Page 3, Last paragraph, line 12 ….Ca2+concentration leads to the increased production of NO, which plays a pivotal role in……. Should there be a space between “Ca2+” and concentration”?        Ca2+ concentration leads to the increased production of NO, which plays a pivotal role in……. It really is not important. If it is difficult to change, it is OK as it appears.
  2. Page 9, para 2, line 16: Thus, the preserved EC integrity may explain the reversal of hypoxia-induced PH on upon return to normoxic conditions (Figure 2).
  3. Page 10, under the subheading “1. Restitution of endothelial caveolin-1:” .the following statement has been added: As shown in Figure 3, cavtratin treatment resulted in slightly less thickening of the pulmonary arteries (A) and preserved endothelial caveolin-1 (B).
  4. Page 10, Figure legend 3A: The following statement has been added. (The pictures were captured at 20x or 40x magnification depending on the size of the arteries).